

# The Founding Actor of Türkiye's Petroleum Geology: Cevat Eyüp Taşman and National Energy Legacy

Oguz Mulayim[1]

[1]Turkish Petroleum Corporation, Adıyaman Directorate, Adıyaman, 02040, Türkiye

*Correspondence to*: Oguz Mulayim (oguzzmlym@gmail.com)

**Abstract.** This article analyses the foundational role of Cevat Eyüp Taşman (1893-1956), Türkiye's first petroleum geologist, by examining how his advanced education in the USA and his professional experience in international oil companies equipped him to become the "*critical human capital*" for Republican Türkiye. Based on archival documents and primary sources, the study scrutinizes the period from his initial engagement in 1929 until his death in 1956, framing his contributions through the

key transformations he spearheaded: the intellectual transformation through systematic field research and scientific publications; the structural transformation via the institutionalization of petroleum exploration within MTA and PDR; and the operational-legal transformation, materialized with the landmark Raman-1 discovery in 1940 and his pivotal contributions to the Petroleum Law No. 6326 in 1954. The research positions Taşman not merely as a technical expert but as a "public intellectual" who integrated scientific knowledge with national development goals. Consequently, his legacy is evaluated on

four foundational pillars—technical, institutional, legal, and intellectual—which collectively underscore his enduring impact on Türkiye's pursuit of energy independence.

## 1. Introduction

The quest for energy independence was a cornerstone of the early Turkish Republic's modernization and political sovereignty. While the 1926 Petroleum Law demonstrated the state's determination, it also revealed a critical deficiency: the absence of

trained local human capital to master the technology and science of oil exploration (Turkish Republic Resmi Gazete, Sayı: 341, 6 Nisan 1926). It was this gap that made the emergence of a pioneer like Cevat Eyüp Taşman (1893-1956) not just beneficial but inevitable for Türkiye's oil destiny (Akcan and İdem, 2023; Akcan, 2024).

This study aims to delineate the portrait of Cevat Eyüp Taşman as the "*founding actor*" of Turkish petroleum geology. By situating his personal journey at the heart of the Early Republic's modernization drive, the article seeks to answer the following

questions:

- How did Taşman's academic formation in the USA and his professional experiences in international oil companies shape his scientific identity?



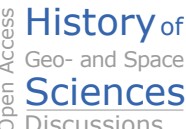

- What intersection of political will and technical necessities led to his invitation back to Türkiye?

- What technical, administrative, and logistical obstacles did he encounter, and how did he overcome them? Furthermore, what setbacks and institutional challenges did he face?

- How did the Raman-1 discovery in 1940 constitute a turning point in Türkiye's energy psychology?

- What personal motivations and worldview drove him to exchange a promising international career for the challenges of a nascent republic?

Using archival documents, this research recasts Cevat Eyüp Taşman from a mere civil servant into a "*public intellectual*"
(Said, 1994) and institution-builder, defined by his scientific acumen, foresight, and patriotism.

## 2. Materials and Methods

This research is based on a critical analysis of archival documents and primary sources. The main archival materials were obtained from the Prime Ministry Republican Archives (Başbakanlık Cumhuriyet Arşivi - BCA) and the Ottoman Archives (BOA), which provided official correspondence, reports, and decisions regarding Taşman's appointment, field operations, and
procurement activities.

Systematic analysis of the Official Gazette (Turkish Republic Resmi Gazete) and Parliamentary Records (TBMM Zabıt Ceridesi and Tutanak Dergisi) formed the basis for understanding the legal and political context. Contemporary newspapers such as Hakimiyeti Milliye, Milliyet, Akşam, and Cumhuriyet were scanned to trace the public perception of oil exploration and Taşman2s role.

Primary sources also include Taşman's own scientific publications, reports, and radio broadcasts, which were essential for analysing his intellectual contributions and public engagement. Memoirs of his contemporaries, notably Kemal Lokman, provided first-hand accounts of the field challenges.

The analysis process involved triangulating these diverse sources to construct a holistic narrative. Thematic analysis was employed to identify and frame Taşman's contributions within the key transformations he spearheaded—intellectual, structural,
and operational-legal—and to evaluate his legacy against the four pillars of technical, institutional, legal, and intellectual impact.

## 3. The Formative Years of Cevat Eyüp Taşman: Academic and Professional Preparation in the USA

The intellectual and professional formation of Cevat Eyüp Taşman (1893-1956) constitutes a paradigmatic example of the "*critical human capital*" that germinated in the late Ottoman Empire and was transferred to Republic of Türkiye (Akcan, 2024).
Excelling in a competitive examination, Taşman was selected as one of five students sent to the United States by the Ottoman

state. His dispatch to Columbia University in 1911 manifested the late Ottoman bureaucracy's resolve to cultivate expertise in modern science and technology (BOA MF. MKT. 1105-31, 14 Ocak 1911; Ayhan, 2021).

The outbreak of the First World War threatened his education with the cessation of state bursaries. This crisis was overcome through intellectual solidarity networks, notably a grant from the Carnegie Foundation, which enabled him to complete his
undergraduate degree in mining engineering in 1915 (Levermore, 1922). Taşman further enhanced his credentials with a master's degree in geology, working as a research assistant to Professor Robert M. Raymond at Columbia University between 1917 and 1918 (Catalogue and General Announcement, 1911-1912; Columbia University Alumni Register, 1932).

The most crucial phase of his preparation was his intensive field experience in the Americas. Over a decade, beginning at the "*Phelps Dodge*" corporation (1918-1920) and continuing at the "*Empire Gas & Fuel Company*" (1920-1930), he conducted
geological surveys and drilling operations in hydrocarbon fields in Texas, New Mexico, and Mexico's Tamaulipas region (The Oil and Gas Journal, June 8, 1933; El Paso Herald, 9 Ekim 1918; Leonard, 1925). This period provided him not only with theoretical knowledge but also with practical experience in tracing oil in challenging terrain.

This nearly twenty-year formative period, from 1911 to 1930, shaped Taşman into a petroleum geologist and field manager of global standards. Armed with this unique combination of academic excellence and practical skill, he was uniquely positioned
to become the most critical actor in Türkiye's quest for oil.

### 3.1 A Forged Identity: Motivation and Worldview

The decision to leave a secure and promising career in the United States was not merely a professional calculation but a reflection of a deeply ingrained worldview. Taşman's formative years, spanning the collapse of the Ottoman Empire and the birth of the Turkish Republic, instilled in him a powerful sense of duty toward national survival and modernization. While
direct personal diaries are scarce, his actions and later writings reveal a figure who saw scientific knowledge not as an end in itself, but as a vital tool for national emancipation.

In a 1949 speech, he reflected this sentiment, stating, 'The greatest service a individual can render is to work for the economic independence of their country, for which scientific and technical knowledge is the most reliable compass' (Taşman, 1949a). This echoes the ethos of the Republic's founding generation, which viewed technical expertise as the bedrock of sovereignty.
His return was thus a conscious choice to align his personal legacy with the project of Turkish nation-building. The challenge of creating something from nothing in his homeland, despite the personal and professional sacrifices involved, appears to have been a more powerful motivator than the prospect of a comfortable career abroad. This fusion of patriotic zeal with scientific rationality became the driving force behind his relentless pursuit of Türkiye's oil goal.



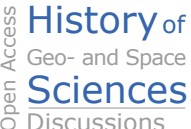

## 4. The Return of Cevat Eyüp Taşman to Türkiye: An Intellectual and Practical Break in National Oil Strategy

In the early Republican period, the ideal of economic independence made national control over energy resources imperative. However, a serious technical and scientific gap, stemming from dependence on foreign experts, hindered this goal. Closing this gap required a paradigm shift in Türkiye's oil strategy. In the pre-Republican era, oil exploration activities were largely limited to the distribution of concessions and very limited geological surveys and shallow drilling carried out by foreign companies in the regions of İskenderun, Thrace, Erzurum-Van, and Mosul, and did not result in an economic discovery (Yalçın,

2024). This period lacked a systematic national petroleum geology framework.

The search for national experts initiated in 1929 under the leadership of the Minister of Economy, Mehmet Şakir Kesebir, was the concrete step for this change. Kesebir's statement that "*the final decision on petroleum exploration activities must be made by Turkish engineers*" reflected the principle of "*full independence*" in energy geopolitics (TBMM ZC, Dönem: 4, İ: 54, C: 2, Cilt: 15, 20.05.1933). The trigger was the journalist Ahmet Emin Yalman's highlighting of his Columbia University classmate,

Cevat Eyüp Taşman, who had accumulated a decade of experience in leading US oil companies (Yalman, 1997). This move signalled the beginning of a transition from the era of foreign consultants to an era of national oil policy led by homegrown experts (Akcan, 2024).

Acting initially as a consultant, Taşman was first invited as a consultant in 1929, marking the start of his advisory role. His definitive return and assumption of a full-time, permanent official capacity followed in 1933, upon the establishment of the

Petroleum Exploration and Operations Administration. The field studies Taşman initiated following his invitation in 1929 constituted the first systematic petroleum geology research of Türkiye's territory in the modern sense. In 1930, an interdisciplinary team comprising Kemal Lokman, the Swiss geologist Dr. Michel Lucius, and the German geophysicist Hoffman conducted geological surveys across a vast geography stretching from Mardin to Thrace (Hâkimiyeti Milliye Gazetesi, 21 Ekim 1930; Milliyet Gazetesi, 10 Nisan 1931). This mobilization symbolized the triumph of scientific

determination under conditions of national infrastructure poverty.

These challenges are strikingly depicted in Kemal Lokman's memoirs: "*In 1930, roads were virtually non-existent in the eastern provinces. We had to conduct our field studies on horseback and on foot*" (Lokman, 1958a). Despite these conditions, the team not only searched for oil but also collected vital data on the region's fundamental geological structure, laying a critical foundation for the geological mapping of Türkiye. These pioneer studies were carried out under extremely difficult physical

and infrastructural conditions—a time when Türkiye had almost no road network, transportation was provided by horse, mule, and on foot, and even basic equipment such as drilling machinery was unavailable (Yalçın, 2024).

Taşman's 1930 report was not merely a field study but also the first comprehensive and analytical strategy document regarding Türkiye's petroleum geology. The report realistically emphasized the "*high-risk, high-cost*" nature of petroleum exploration and, considering the state's limited budget, offered a pragmatic proposal to direct private capital into this field in the initial

phase. This demonstrated that Taşman was not only a geologist but also a strategist.


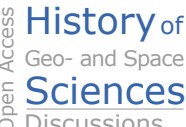

Even more importantly, Taşman's 1931 article titled "*Petroleum Possibilities of Türkiye*", published in the "*Bulletin of the American Association of Petroleum Geologists*" one of the world's most respected geology journals (Taşman, 1931), was the first academic reference to systematically present Türkiye's oil potential to the international scientific community. This publication laid the intellectual foundation of Turkish petroleum geology in global literature and served the strategic function

of attracting the attention of foreign investors and companies (The New York Times, 6 July 1925).

The work of Cevat Eyüp Taşman between 1929 and 1931 built a lasting intellectual legacy by making three fundamental contributions to Türkiye's national oil strategy:

- Mindset Transformation: It represented the symbolic beginning of the shift from leaving technical decisions to foreign experts to building national expertise.

- Methodological Revolution: It implemented systematic geological survey and mapping methodology in Türkiye for the first time.

- Global Promotion and Legitimization: It positioned Türkiye as an "exploration region" in petroleum geology literature, granting it scientific legitimacy.

This period laid the intellectual and scientific groundwork for Türkiye's energy independence struggle, paving the way for

Taşman's formal leadership in 1933.

**4.1 Taşman in Comparative Perspective: "Critical Human Capital" in the Global Energy Landscape**

The early 20th century witnessed a global scramble for oil, necessitating the development of technical expertise worldwide. The concept of" critical *human capital* "was not unique to Türkiye, but its manifestation in Taşman's profile reveals a distinct trajectory. Comparing his role with contemporaries in other regions illuminates the specificities of the Turkish experience. In

the Soviet Union, a similar drive for technological autarky created figures like Ivan Gubkin, a geologist who founded the Soviet petroleum geology school and led exploration in the Volga-Ural region. Like Taşman, Gubkin was a scientist-bureaucrat who integrated research, education, and state planning. However, Gubkin operated within a rigid, state-controlled ideological framework, whereas Taşman navigated a nascent republic seeking to blend étatisme with engagement with Western capital and technology (Kontorovich, 2017).

In the Middle East, the pattern was markedly different. In countries like Iran and Iraq, oil exploration was dominated by foreign concessionary companies (e.g., Anglo-Persian Oil Company). The initial" critical "expertise was almost entirely imported, with limited knowledge transfer to a local cadre until much later (Yergin, 2003). Taşman's case is a powerful contrast: as a nationally rooted expert who had mastered foreign science and field practice, he returned not as an employee of an international cartel, but as a servant of the state, aiming to build indigenous capacity from the outset.

This comparative glance underscores Taşman's paradigmatic uniqueness. He was neither a product of a closed, ideological system like his Soviet counterparts, nor a latecomer in a peripherical economy dominated by foreign corporations. He represented a hybrid model: a "*Western-trained national expert*" whose authority was derived from his international credentials

and practical experience, yet was deployed with the explicit goal of national emancipation in the realm of strategic technology. This positioned him as the chief architect of a uniquely Turkish path to building petroleum sovereignty.

A comparison with contemporary "*late-developer*" nations reveals further nuances. In Latin American countries like Mexico and Venezuela, the model often involved initial foreign domination followed by a forceful nationalization driven by political and popular movements, with technical expertise catching up later. Taşman's Türkiye presented a distinct case: a preemptive build-up of national technical sovereignty prior to large-scale foreign entry, with Taşman as its chief architect.

Furthermore, Taşman's profile as a "*Western-trained national expert*" was part of a broader pattern in the Early Republic,
visible in figures like railway engineer Behiç Erkin or aviation pioneer Vecihi Hürkuş. These were individuals who internalized foreign technology and expertise but deployed it with an uncompromisingly nationalistic vision. Taşman's uniqueness within this cohort lies in the strategic, long-term institutional and legal frameworks he established, which outlasted his tenure and shaped the industry for decades.

## 5. Institutional Construction, Operational Breakthrough, and Legal Framing (1933-1955)

This period, encompassing Taşman's formal leadership, marks the transformation of Türkiye's oil ambitions from disparate initiatives into a structured, state-led industry. It was characterized by the establishment of foundational institutions, a landmark discovery, and the creation of a modern legal framework, all under his strategic guidance (Akcan, 2024).

### 5.1 Foundation and Early Exploration (1933-1935)

Although the 1926 Petroleum Law demonstrated the state's will, it suffered from a lack of a central technical authority. The
Petroleum Exploration and Operations Administration, established by Law No. 2189 on May 20, 1933, was the definitive response (Resmi Gazete, 27 Mayıs 1933, s. 2508). This institution embodied the idea of national energy independence.

Building on his earlier contributions as a special advisor, Taşman accepted the Directorship on May 29, 1933, leaving behind his career in the USA (BCA, 30-18-01-02/36-40-20, 29 Mayıs 1933; Hâkimiyeti Milliye Gazetesi, 3 Haziran 1933; Milliyet Gazetesi, 25 Nisan 1933). His vision was to create a "*research and application center*". His first strategic move was to form
an interdisciplinary team, bringing world-renowned geologists Sidney Paige and Harold Moses to Türkiye (Milliyet Gazetesi, 17 Nisan 1933; Milliyet Gazetesi, 12 Temmuz 1933; Vakit Gazetesi, 20 Eylül 1933) (Figure 1). The intensive 70-day field work that began on July 12, 1933, aimed to create a detailed inventory of Southeastern Anatolia using modern geological and geophysical methods, elevating exploration from "*tracking traces*" to "*scientific prediction*".

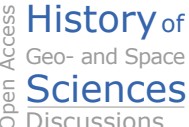
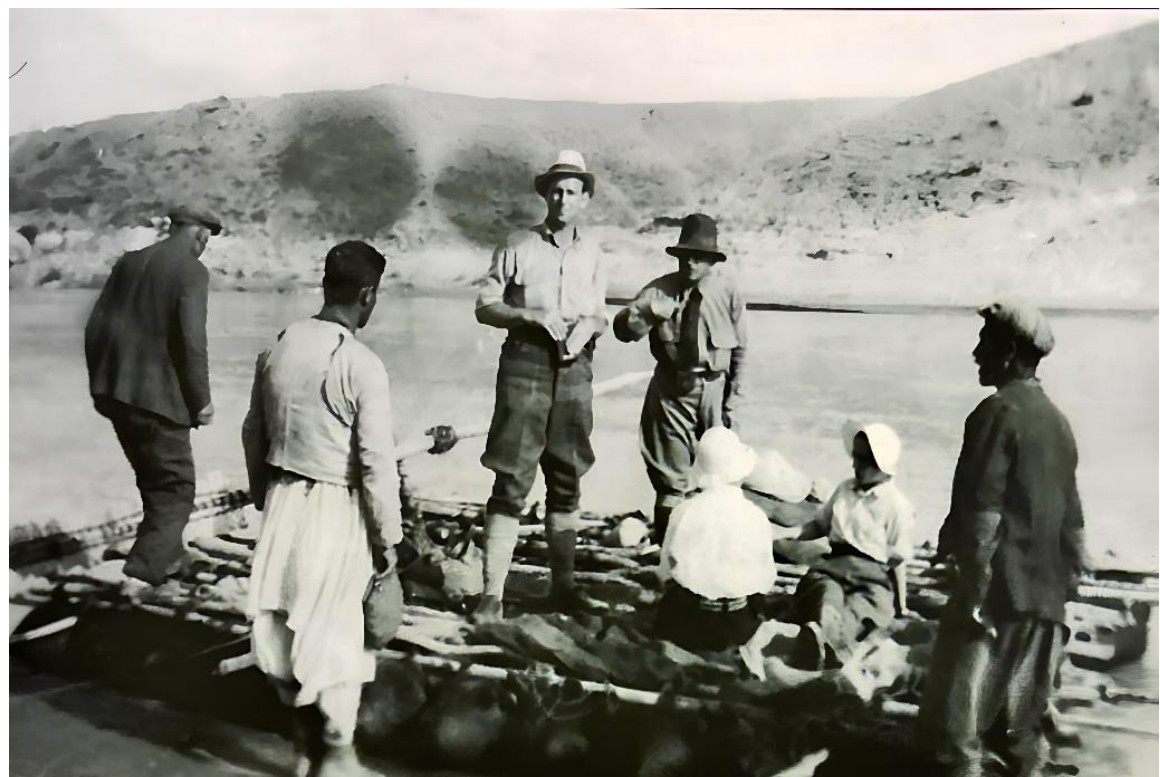

**Figure 1: Harold Moses and Cevat Eyüp Taşman during geological field surveys in Southeastern Anatolia, 1937. This image captures the collaborative international expertise that Taşman integrated into Türkiye's early petroleum exploration efforts, symbolizing the shift from foreign consultancy to nationally directed scientific research (from, Özcan, 2006).**

The drilling of the Basbirin-1 well in the Raman region on October 13, 1934, became a symbol of scientific courage (Akşam Gazetesi, 6 Haziran 1934; Yeni Asır Gazetesi, 15 Ekim 1934). Although it did not find commercial oil, it was Türkiye's first modern deep oil exploration drill. Basbirin-1 conquered the "*fear of not finding oil*" and charted the course for the future Raman-1 discovery (Figure 2). The Basbirin-1 well, drilled with this second-hand steam-powered percussion machine, marked the beginning of modern deep oil exploration drilling in Türkiye. Although it resulted in a 'dry hole' at 1,327 meters, it was

regarded as a test of and a boost to the country's technical capacity and self-confidence (Yalçın, 2024). This initial phase established a lasting strategic legacy based on a Structural Transformation (a central, professional institution), a Technological Transformation (adoption of international standard methodologies), and a Human Capital Transformation (training Türkiye's first generation of modern petroleum engineers).



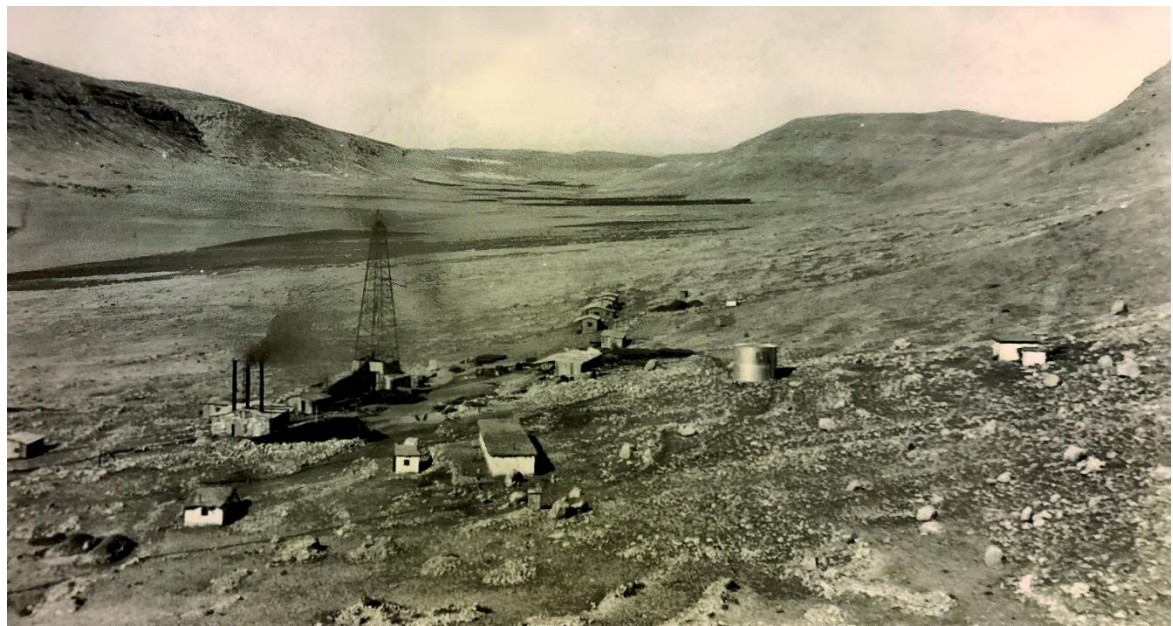

**Figure 2: The Raman-1 drilling site at Maymune Gorge, 1940. This well, which struck oil at 1048 meters, marked Türkiye's first major petroleum discovery and became a symbol of national resilience and technical achievement during World War II-era resource constraints (from Özcan, 2006).**

However, this initial phase was not without its setbacks and internal challenges. The strategic focus on the Raman region,
which would later prove successful, came only after several unsuccessful and costly drillings elsewhere, particularly around Mardin. These early failures, while not extensively documented in public reports, highlighted the inherent high-risk nature of petroleum exploration and the limitations of the existing geological data. Taşman faced skepticism from within the bureaucracy regarding the high costs of deep drilling, with some officials questioning the return on investment in a largely unexplored geological terrain. His ability to secure continued funding and political support for the Raman project, despite these initial
disappointments, was a testament to his credibility and strategic persuasion.

### 5.2 The Raman-1 Discovery and Wartime Resolve (1939-1945)

The establishment of the Mineral Research and Exploration Institute (MTA) in 1935 consolidated mining and petroleum activities under a scientific authority (Resmi Gazete, 22 Haziran 1935, s 5378). Taşman's appointment as head of the Petroleum
Group Directorate signalled the definitive transition to"national will and expertise".





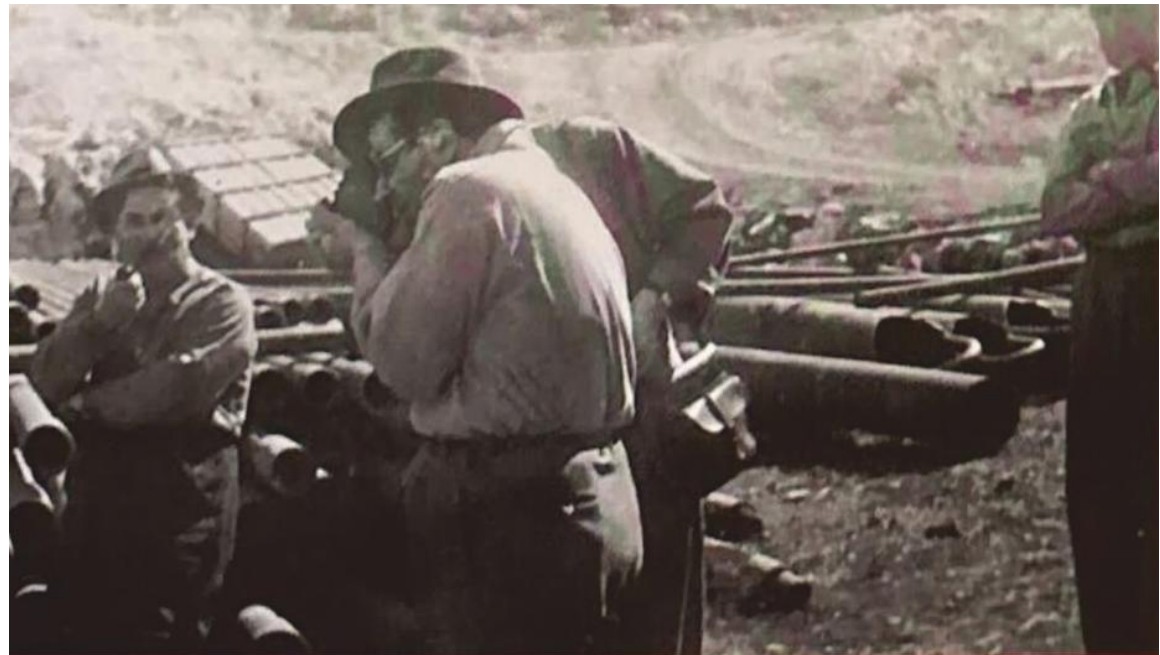

**Figure 3: Cevat Eyüp Taşman (right) with an American driller at a field site in the late 1930s. The presence of foreign technical personnel underscores the transfer of advanced drilling technology and know-how, which Taşman strategically facilitated to build local capacity (from Özcan, 2006)**


The pinnacle of this era was the 1940 Raman-1 discovery. After unsuccessful drillings around Mardin (Figure 3). Taşman refocused strategy on the Raman Mountain region, overcoming extraordinary logistical challenges, including a 350km detour for equipment due to the absence of a bridge over the Tigris (Figure 4) (Lokman, 1940; Yeni Sabah Gazetesi, 22 Mayıs 1940). On April 20, 1940, amidst World War II resource constraints, oil was struck at 1048 meters. This was not merely a technical
success but a monumental victory for national morale, proving that energy independence was achievable. Following the dry or non-commercial results of the other wells in the region (Raman-2 through 7) after Raman-1, a new technical evaluation committee chaired by Necdet Egeran was established within the MTA. With a strategic decision, the Raman-8 (Sirmen-1) well, drilled at a different location on the anticline, achieved Türkiye's first commercial oil discovery in January 1946. This was followed by the discovery of the Raman-9 well in 1948 and shortly after by the discovery of the Garzan field (Yalçın,
2024). In this process, İhsan Ruhi Berent's administrative support and Necdet Egeran's technical leadership demonstrated that the institutional foundation laid by Taşman was bearing its first fruits.

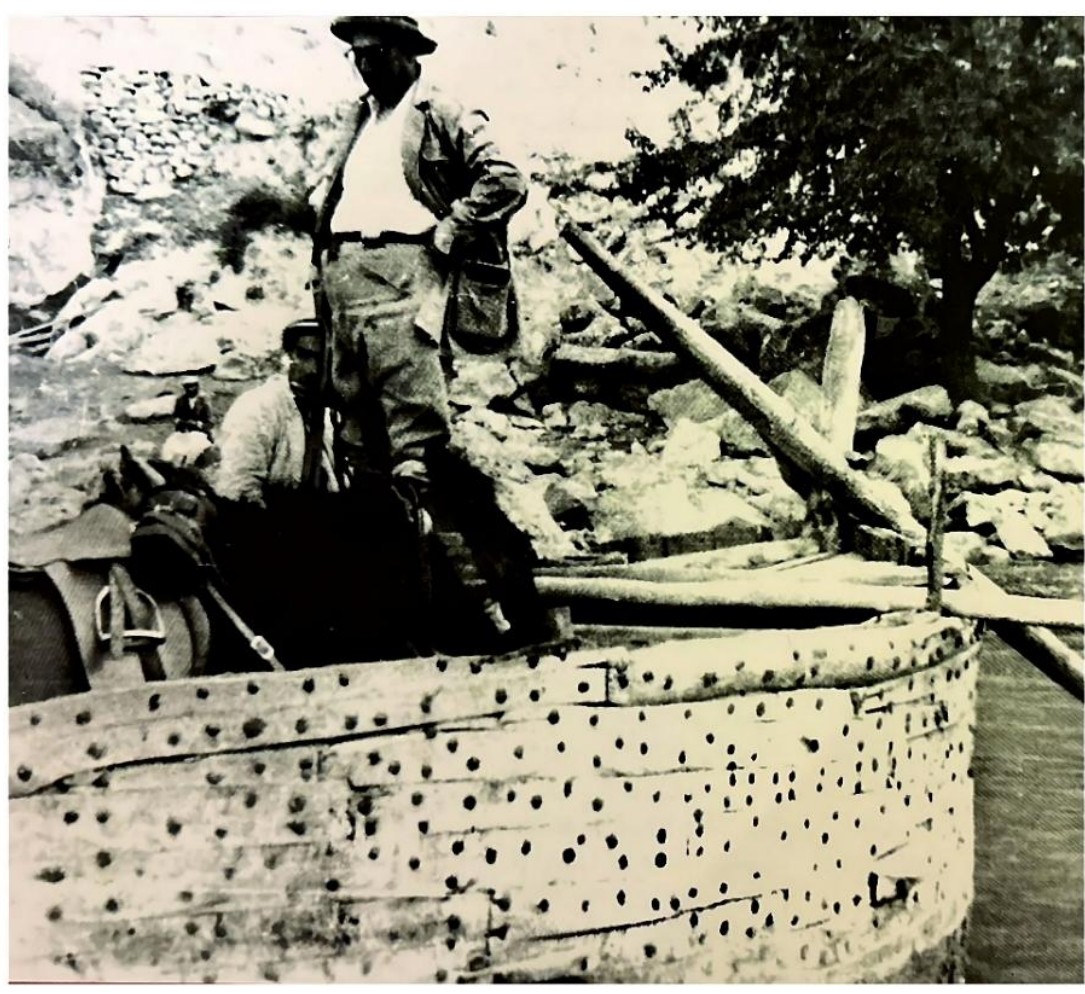

**Figure 4: Cevat Eyüp Taşman crossing the Tigris River on a kelek (traditional raft), circa 1940. This image illustrates the extreme logistical challenges faced by early exploration teams in remote Eastern Anatolia, where lack of**
**infrastructure necessitated improvisation and perseverance (from Özcan, 2006).**

Taşman demonstrated further adaptability by procuring modern Rotary drilling systems from Iraq in July 1940, revolutionizing operational efficiency (BCA, 30-18-01-02/92-74-7, 23 Temmuz 1940). Taşman's efforts during this challenging period were supported by valuable contributions from his wife, Mehlika İzgi Taşman-Ribnikar (1912–2007), Türkiye's first female
petroleum geologist, both in the field and in the laboratory. Mehlika Taşman's interest in geology began with her education at the American College for Girls (ACG) in Istanbul. Sent by the state in 1938 to the United States for geological training, she completed her master's degree at the University of Texas at Austin, a hub for micropaleontology. Under the guidance of her advisor and former teacher, the renowned micropaleontologist Dr. Louise Jordan, she conducted her thesis work titled "*A Study of the Micro-Fauna Related to the Adana Drilling Operations*" (Izgi, 1940; Okay, 2019).




She was urgently recalled to Türkiye at the height of World War II, during preparations for the opening of the Raman-8 (Sirmen-1) well. After a difficult journey, she returned to Türkiye in 1941 and was immediately assigned as the well-site paleontologist for the Garzan-1 well in the Raman field. There, she performed micropaleontological analysis of drill cuttings, enabling stratigraphic correlation and providing a critical contribution to the understanding of the region's subsurface geology

(Özcan, 2006; Taşman-Ribnikar, 1975). Her detailed foraminiferal analyses on samples from wells in the Raman and later the Adana basins formed the basis for the correlation of wells, particularly the Hocalı and Ağzikara wells, and for interpreting the regional paleogeography (Taşman 1957a, 1957b).

Her career continued to be pioneering after she left MTA. She worked for Esso Türkiye, established a private micropaleontology laboratory, and spearheaded the establishment of Turkish Petroleum Corporation's (TPAO) first

micropaleontology laboratory in Batman in 1957. To pass on her knowledge, she organized "*Applied Micropaleontology*" courses at Istanbul Technical University and published her lecture notes as a book under the same title in 1975 (Taşman-Ribnikar, 1975). This work became a fundamental guide for subsurface geology laboratories in Türkiye. Actively involved in professional organization, Mehlika Taşman was among the founding members of the Geological Society of Türkiye (TJK) in 1946 and later served as the President of the Turkish Association of Petroleum Geologists, achieving an esteemed position

within the community as a female scientist (Okay, 2019, 2021).

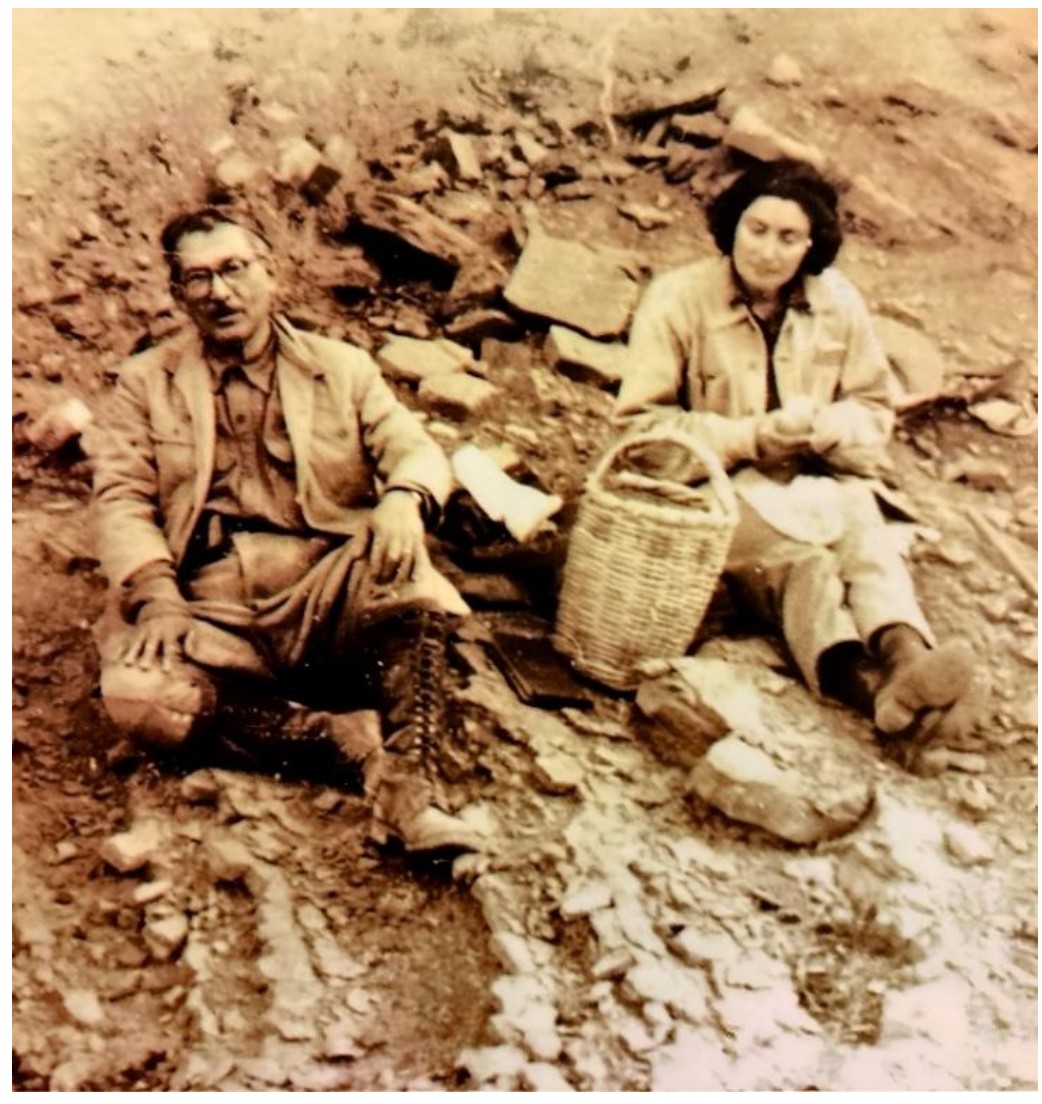

**Figure 5: Cevat Eyüp Taşman with his wife, Mehlika İzgi Taşman-Ribnikar, during fieldwork. Mehlika, Türkiye's first female petroleum geologist, earned her MSc at the University of Texas at Austin under the mentorship of micropaleontologist Dr. Louise Jordan. She served as the well-site paleontologist for critical wells such as Garzan-1 and conducted pioneering quantitative foraminiferal analyses for the correlation of the Adana basin wells. Her legacy extends to founding TPAO's first micropaleontology lab and authoring the foundational handbook "Applied Micropaleontology" (from Özcan, 2006).**

His mission extended beyond the field. His 1944 Ankara Radio program, "*The Primary and Raw Materials of This Era: Petroleum*", and his 1946 public lecture at Ankara University, "*What is Petroleum and How is it Sought*?", cemented his role





as a "*public intellectual*", demystifying petroleum geology and instilling its strategic importance in the public consciousness (Taşman, 1944; Ulus Gazetesi, 3 Ağustos 1937; Tan Gazetesi, 22 Mart 1937; Tan Gazetesi, 7 Ağustos 1937).

The institutional journey within MTA was also marked by challenges. As the organization grew, Taşman had to navigate bureaucratic rivalries and competition for limited resources between the petroleum group and other mining divisions. His

leadership was tested not only in the field but also in Ankara's corridors of power, where he had to consistently argue for the strategic priority of petroleum exploration over other pressing national needs, especially during the austerity of the war years. The procurement of the Rotary drill from Iraq, while a tactical masterstroke, was itself a bureaucratic feat, requiring overcoming logistical red tape and international trade restrictions in a war-torn region.

### 5.3 Legal Framing and Institutional Legacy (1950-1955)

By the 1950s, the outdated Petroleum Law No. 792 necessitated a new legal framework. Taşman personally led the technical committee for over a year and a half, ensuring the new Petroleum Law No. 6326 was both legally sound and operationally pragmatic (TBMM TD, Dönem:27, Yasama Yılı:2, Cilt:11 B:48, 5.02.2019). Enacted in 1954, it opened the way for private sector investment and integrated Turkish petroleum with global standards (Resmi Gazete, 16 Mart 1954, s.8633-8643).

The new law required a new regulatory body, leading to the establishment of the Petroleum Affairs Directorate (PDR) in 1954

(Resmi Gazete, 22 Mart 1954, s. 8773-8774). Taşman's transfer to the PDR as a Technical Advisor in 1955 signified the transfer of his field experience to the core of the state's regulatory mechanism. The Petroleum Regulations, enacted under his guidance in 1955, completed the legal infrastructure (Petrol Dairesi, 1957). Thus, Taşman uniquely helped found both the exploration company (MTA) and the regulatory body (PDR).

The transparent climate created by Law No. 6326 attracted immediate international interest, with 187 license applications in

1955 alone—a testament to confidence in both the new legal framework and Taşman's undisputed technical authority (MTA, 1955). Thanks to the transparent and encouraging environment created by Petroleum Law No. 6326, the number of domestic and foreign oil companies operating in Türkiye reached 19 in 1957, exploration activities accelerated, and Türkiye's oil production basins diversified with the discovery of new fields such as Germik in 1958, Bulgurdağ (the first light oil) in 1960, and Batı Raman and Kayaköy in 1961 (Yalçın, 2024).

This twenty-year leadership period built the Turkish petroleum industry on four fundamental axes:

- Institutional Identity: Establishing a professional, autonomous structure within the MTA and PDR.
- Technical Autonomy: Achieving the Raman-1 discovery and adapting modern technologies.
- Legal Framework: Providing the technical leadership for Petroleum Law No. 6326.
- Intellectual Legacy: Fostering public awareness and establishing petroleum geology as a "*development ideal*".



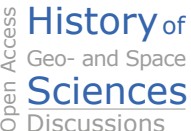

**6. Institutional Building and Intellectual Leadership: Taşman's Professional Associations and Academic Legacy**

Cevat Eyüp Taşman's professional identity was shaped in a arena far broader than his office or the field. One of the most enduring marks of this identity was his role as one of the five founding members of the Geological Society of Türkiye on July 14, 1946, alongside names such as Hamit Nafiz Pamir, Recep Egemen, Cahide Kırağlı (Ünsalaner), and Necdet Egeran (Bayramgil, 1954). Notably, his wife, Mehlika İzgi Taşman, also stood among these pioneering founders, marking her

institutional role alongside her husband (Okay, 2021). This initiative was an intellectual mobilization aimed at gathering dispersed geologists under a common roof and establishing professional solidarity and ethical standards. Taşman's role here was that of an "intellectual leadership" that institutionalized the idea that the country's subsurface resources must be investigated on scientific foundations (Ketin, 1985).

Taşman's influence within the institution was not limited to founding membership. His service as the President of the

Geological Society of Türkiye for the 1949-1950 (4th Term) and 1950-1951 (5th Term) years proved that he was a respected science manager. These two terms focused on vital issues such as ensuring the society's permanence, encouraging periodical publications, and institutionalizing professional training activities. Taşman's leadership served as a catalyst for the Turkish geology community to evolve from a network into an effective mechanism for producing science and proposing policy (Ercan, 1985).

The most tangible outputs of Taşman's intellectual legacy are his impressive list of publications, comprising nearly 30 scientific articles and congress papers (Taşman, 1931; 1944; 1949; 1954). His 1948 congress paper (Taşman, 1948a) presented the scientific findings from the Raman-1 discovery (1940) to the international community. These works reflect his role in two different arenas:

*A Representative in the Global Arena:* His publications in the world's most respected geology journals and platforms, such as

the American Association of Petroleum Geologists (AAPG) Bulletin and International Geological Congress (IGC) reports, made him a standard-bearer in the international scientific community (Erguvanlı, 1980). His articles announced Türkiye's geological potential to the world while contributing to universal science.

*An Educator and Guide in the Local Arena*: His articles in national publications, such as the MTA Institute Journal, aimed to share his accumulated knowledge within the country, train young geologists, and contribute to the enrichment of Turkish

geology literature (Taşman, 1949).

Cevat Eyüp Taşman's legacy in the context of professional associations and academic productivity rests on three fundamental, intertwined pillars (Akcan, 2024);

- **Professional Organization**: Ensuring the profession attained an organized structure as a founding member of the Geological Society of Türkiye.

- **Scientific Management**: Guiding the Geological Society as President, ensuring institutional continuity and revitalizing scientific platforms.

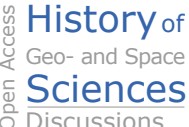

- **Knowledge Production:** Producing scientific publications that positioned Türkiye as a" subject "in geological literature and paved the way for training domestic scientists.

This multi-layered approach made Taşman not only a geologist who found oil, but also a founding actor in Türkiye's integration
with modern science and a pioneering "*public intellectual*" figure.

## 6.1 Taşman as a "Public Intellectual": Bridging Science and Society

The analysis of Taşman's legacy would be incomplete without a deeper conceptualization of his role as a "*public intellectual*". Drawing on Edward Said's seminal definition, the public intellectual is not merely a specialist, but a figure who "*speaks truth to power*"and utilizes their knowledge to address and enlighten a broader public audience, translating specialized discourse
into matters of common concern (Said, 1994). Taşman embodied this role quintessentially.

His scientific publications established his technical authority, but it was his concerted efforts to democratize knowledge that cemented his status as a public intellectual. His 1944 radio broadcast, "*The Primary and Raw Materials of This Era: Petroleum,*" on Ankara Radio was a deliberate act of public pedagogy. In an era of limited mass communication, he chose the airwaves to explain the strategic significance of oil, moving the discourse from closed technical committees into the living
rooms of citizens. Similarly, his 1946 public lecture at Ankara University, "*What is Petroleum and How is it Sought?*", was not an academic seminar but an accessible guide aimed at demystifying the science for students and the educated public.

Through these acts, Taşman performed the essential function of the public intellectual: he mediated between the specialized world of petroleum geology and the normative realm of national development. He did not merely implement state policy; he created a web of public understanding and support for it, instilling a "petroleum consciousness"as a component of modern
citizenship. Taşman's public awareness efforts gain even greater significance considering that, during this same period, petroleum geology education had not yet been introduced at universities. Independent geology undergraduate programs in Türkiye began only in 1946, and petroleum geology courses were not included in the curriculum until the 1960s. For this reason, Taşman's radio talks and popular articles long served as the primary source of public and student knowledge in this field (Yalçın, 2024). This aligns with Said's view of the intellectual's role in raising popular consciousness. By making complex
geological and economic concepts accessible, he empowered the public to grasp the "national cause" of energy independence, thereby building a social foundation for the technical and institutional structures he helped create. This synthesis of elite expertise and public engagement distinguishes him from a purely technical bureaucrat and positions him as a foundational figure in the republic's scientific and intellectual history.

The impact of these public engagements was tangible. Following his radio broadcast, newspapers reported a noticeable increase
in public inquiries to the MTA regarding petroleum and mineral resources, indicating a success in sparking public curiosity (Cumhuriyet Gazetesi, 15 Mayıs 1944). More concretely, his lectures and articles are credited by contemporaries with inspiring

**History** of
Geo- and Space
**Sciences**
Discussions

a new generation of Turkish students to pursue geology and mining engineering. Prominent geologists who emerged in the 1950s often cited Taşman's public talks and accessible writings as a key influence in their career choice (Ercan, 1985).

This was not a one-way transmission of knowledge but the cultivation of a social contract for science. By demystifying petroleum geology, Taşman was building a crucial layer of public legitimacy and support for the state's technically complex and financially demanding energy projects. He transformed the abstract concept of 'energy independence' into a relatable national goal, creating a web of understanding that fortified the technical and institutional structures he helped build against potential political and economic headwinds.

## 7. Conclusion

This research set out to illuminate the foundational role of Cevat Eyüp Taşman in Turkish petroleum geology by probing key questions about his formation, return, challenges, achievements, and motivations. The findings provide clear answers and position his legacy within broader academic discourses.

First, Taşman's scientific identity was forged through a unique synthesis of elite academic training at Columbia University and extensive practical field experience in the US oil industry. This combination equipped him with not only technical prowess

but also a strategic, field-tested perspective rare in his homeland.

Second, his invitation back to Türkiye was precipitated by a critical convergence of political will—embodied in Minister Mehmet Şakir Kesebir's push for "full independence"—and a stark technical necessity, highlighted by Ahmet Emin Yalman. The young Republic needed an expert who could bridge the gap between international standards and national ambitions.

Third, upon his return, Taşman confronted immense obstacles: a near-total lack of infrastructure, bureaucratic skepticism,

limited budgets, and early exploratory failures around Mardin. He overcame these through a combination of scientific rigor (pioneering systematic surveys), strategic pragmatism (refocusing on Raman), diplomatic skill (navigating bureaucracy to procure essential equipment), and unwavering resolve.

Fourth, the Raman-1 discovery in 1940 was far more than a technical success. It was a profound psychological turning point that shattered a national inferiority complex regarding oil and embedded a foundational confidence in the nation's capability

to achieve energy independence.

Finally, his decision to leave a promising career abroad was driven by a worldview that fused patriotic duty with a belief in scientific progress as the bedrock of national sovereignty. He saw his knowledge not as a personal asset but as a tool for national emancipation, aligning his personal legacy with the Republic's nation-building project.

Taşman's true legacy, therefore, lies not in a single discovery but in the robust ecosystem he architected. He was instrumental

in building the technical, institutional (MTA and PDR), legal (Petroleum Law No. 6326, which modernized the regulatory framework and attracted international investment), and intellectual pillars of the Turkish petroleum industry. His untimely death in 1956 marked the end of a formative era, but the technical, institutional, and legal foundations he had laid proved resilient. His legacy lived on through the continued work of MTA and PDR, the thriving professional community he helped

establish, and the enduring exploration mindset he instilled in the nation's energy sector. The subsequent discoveries in the following decades stand as a testament to the robustness of the ecosystem he architected. His career exemplifies a unique model of the "*state-employed public intellectual*", who seamlessly translated specialized knowledge into public awareness and institutional reality.

This study contributes to several scholarly fields. In the history of Turkish technocracy, it offers a granular case study of how "critical human capital" was operationalized to achieve technological sovereignty. From a sociology of science perspective, it
demonstrates how a professional scientific community and its ethics were institutionalized in a late-developing country, significantly through Taşman's leadership in the Geological Society of Türkiye. For energy geopolitics, it presents a distinct, hybrid Turkish path that strategically blended étatisme with controlled engagement with foreign capital and expertise, differing from both the Soviet state-control model and the Middle Eastern concessionary model.

Future research could build on this foundation. Comparative studies with other Turkish technocrats in different sectors could
further illuminate the Early Republic's unique model of expertise. Deeper analysis of the public reception of Taşman's popular science efforts would enrich our understanding of science-society relations in the period. Finally, his strategies for negotiating technology transfer and foreign investment offer valuable historical lessons for contemporary debates on resource nationalism and global energy partnerships. The system established by Taşman continued its course in the subsequent period with new discoveries such as Gabar Mountain (2022) and deep-sea successes like the Sakarya Gas Field in the Black Sea (2020).
However, due to Türkiye's complex geological structure and petroleum systems consisting of small-to-medium scale fields, domestic production has historically been able to meet only 8-10% of the demand (Yalçın, 2024). This situation reaffirms that the ideal of 'energy independence,' which Taşman foresaw and fought for, cannot be achieved without technical autonomy and persistent exploration determination.

In an era where energy security is paramount, Taşman's synthesis of patriotic dedication, scientific rigor, strategic institution-
building, and public education remains a powerful compass. His story underscores that the most critical resource for a nation's destiny is not merely what lies beneath the ground, but the qualified human capital and unwavering will to develop it.

**Appendix A: Cevat Eyüp Taşman – publications 1931–1956**

Taşman, C. E.: Petroleum Possibilities of Turkey, B. Am. Assoc. Petrol. Geol., 15, 629–681, 1931.
Taşman, C. E.: Mürefte'de Petrol Aramaları (Search for Oil in Mürefte), MTA Mo., 3, 1936a.

Taşman, C. E.: Van Gölü Civarında Korzot Petrolü (Oil at Korzot near Lake Van), MTA No., 5, 1936b.

Taşman, C. E.: Türkiye ve Petrol (Turkey and Petroleum), MTA No., 3, 1937a.

Taşman, C. E.: Orta Anadolu'nun Tuz Domları (Salt Domes of Central Anatolia), MTA No., 4, 1937b.

Taşman, C. E.: Trakya Jeolojisi Hülâsası ile Trakya Petrol Aramaları Durumu (A Geologic Synopsis and Status of Oil
Exploration in Thrace), MTA No., 3, 1938a.

Taşman, C. E.: Petrol Aramaları -1923 den Evvel ve Sonra, MTA No., 4, 1938b.

Taşman, C. E.: Cenubi Türkiye'de Petrol ihtimalleri (Oil Possibilities in Southern Turkey), MTA No., 2, 1939a.



Taşman, C. E.: Petrol Aramaları ve Bulma İmkânları, MTA No., 4, 1939b.

Taşman, C. E.: Adana Petrol Sondajının Hususiyeti (Some Particulars about Drilling at Adana), MTA No., 2/19, 1940a.

Taşman, C. E.: Petrol Bulma İmkânları, MTA No., 4/21, 1940b.

Taşman, C. E.: Petrol Bulunması, MTA No., 2/27, 1942.

Taşman, C. E.: Gerede-Bolu Depremi, MTA No., 1/31, 1944a.

Taşman, C. E.: Bu Devrin Ana ve Ham Maddeleri: Petrol, MTA No., 2/32, 303–307, 1944b.

Taşman, C. E.: Tuzlarımız, MTA No., 1/33, 1945a.

Taşman, C. E.: Trakya ve Petrol (Thrace and Oil), MTA No., 2/34, 1945b.

Taşman, C. E.: Harbolit-Kömürlü bir Asphalt (Harbolite: A Carbonaceous Hydrocarbon), MTA No., 1/35, 1946a.

Taşman, C. E.: Varto ve Van Depremleri, MTA No., 2/36, 1946b.

Taşman, C. E.: The Stratigraphy of the Alexandretta Gulf Basin, in: Report of the Eighteenth Session, Great Britain, Part VI, International Geological Congress, 1948a.

Taşman, C. E.: Türkiye Cenup-Doğu Bölgeleri Stratigrafisi, MTA No., 38, 1948b.

Taşman, C. E.: Petrolün Türkiye'de tarihçesi, Maden Tetkik ve Arama Dergisi, 39, 14–22, 1949a.

Taşman, C. E.: Drilling for Oil in Turkey, Oil Forum, 1949b.

Taşman, C. E.: Türkiye'de Bitümlü Tezahürlerin Stratigrafik yayımı, MTA No., 40, 1950.

Taşman, C. E.: On the Oil Possibilities of Turkey with Special Reference to the Raman Field, in: Proceedings of the Third
World Petroleum Congress, The Hague, Netherlands, 1951.

Taşman, C. E.: Petrol aramalarında Stratigrafinin Önemi, Türkiye Jeoloji Kurumu Bülteni, IV, 1, 1953.

Taşman, C. E.: Turkey's Oil Prospects Inviting, The Petroleum Engineer, 26, 9, 1954.

Taşman, C. E.: Evidences of Oil and Gas Associated with Igneous Rocks in Turkey, in: Proceedings of the XX International Geological Congress, Mexico City, Mexico, 1956.

**Data availability**. No data sets were used in this article.

**Author contributions**. This paper was conceptualized by OM. The investigation, writing and editing were performed by OM.

**Competing interests**. The author has declared that there are no competing interests.

**Acknowledgements.** The author would like to thank the Turkish Petroleum Corporation (TPAO) for their support. I am
appreciative of Fatih Köroğlu insightful comments, edits and suggestions regarding the text.

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
