# Peer review of "The Founding Actor of Türkiye's Petroleum Geology: Cevat Eyüp Taşman and National Energy Legacy"

_History of Geo- and Space Sciences, 2026_

## Referee Comment (RC2)

**Response to Author Comments**

I thank the author for the response and for the revisions made to the manuscript. The corrections and additions are appreciated and have contributed positively to the manuscript.

Below are my comments regarding the present version of the manuscript:

1. I am pleased to see that the author has made a serious effort to revise the structure, language, and citations. Unfortunately, I was unable to locate or download the revised version of the manuscript. In other words, I could not access the updated text as a PDF reflecting these changes. Reviewing the revised manuscript itself would have been considerably more helpful than evaluating the revisions in the abstract.

2. Although I still do not agree with the preference for using "Türkiye" instead of "Turkey," I understand the author's concern, particularly given his status as a public official and the possible personal risks associated with this choice. I therefore leave this issue to the discretion of the editorial board. If the editorial board approves the use of "Türkiye," the author may retain this usage consistently throughout the manuscript.

3. In Line 399, the author disagrees with my statement that the natural gas prospects of the Black Sea were known decades ago. One illustrative example which endorses this fact is:

   *Aydemir, V., Sefunç, A., Abalıoğlu, İ., & Efeoğlu, T. (2001). Hydrocarbon potential of Western Black Sea Offshore, Turkey. Proceedings of the 13th International Petroleum Congress and Exhibition of Turkey (June 4–6, 2001, Ankara), pp. 14–20.*

   Just like the Gabar Oil Field, the Sakarya Gas Field is merely the most recent outcome of previously known and ongoing exploration activities. Yet the manuscript is framed in a way that suggests an absence of exploration efforts from Taşman's death until the present government, implying that his legacy has only now been realized—an implication I do not agree with. In my assessment, the recent investments in the Sakarya Gas Field and the Gabar Oil Field have no direct relation to Taşman's dedicated and patriotic efforts during the early years of Kemalist Turkey. From the most straightforward perspective, the severe public financial losses associated with these investments would not have been tolerated by individuals with Taşman's patriotic principles.

   If the author nevertheless insists on linking Taşman directly with these investments, this must be clearly stated as the author's own interpretation.

4. In Line 401, if the author intends to state that Türkiye meets 9.8% of its hydrocarbon demand from domestic resources, this claim must be supported by an appropriate reference, such as the "TPAO Sector Report 2024", which should be cited in the text and added to the reference list.

5. In Lines 402–403, the author explains that "*geology is the primary constraint in petroleum exploration*", "*no amount of technology, investment, or human determination can create commercial hydrocarbon reserves where suitable source rocks, reservoirs, traps, and seals do not exist*", and "*Turkey's overall petroleum geology is challenging compared to global super-basins*". **This is precisely the point I was making in my earlier comment**. However, the

current wording in the manuscript implies that Turkey has significant hydrocarbon potential to meet its own needs but only lacks the necessary technology or determination to exploit it. While I appreciate the author's enthusiasm for reminding me of basic petroleum geology, the only revision required here is a correction of the misleading implication created by the current wording (the sentence needs to be rephrased to convey the correct meaning).

I thank the author once again for preparing this important manuscript on Cevat Eyüp Taşman. I would like to see the revised final version of the manuscript, and I request that the ambiguity in Lines 399–403 be resolved.

---

## Author Comment (AC1)

**Response to Reviewer Comments**

I sincerely thank the reviewer for their careful reading and constructive feedback on my manuscript, "The Founding Actor of Türkiye's Petroleum Geology: Cevat Eyüp Taşman and National Energy Legacy." I appreciate the reviewer's acknowledgment of Taşman's significance and have found the detailed critiques highly valuable for strengthening the paper. I have carefully considered each point and outline below our planned revisions for a major restructuring and refinement of the manuscript

**Reviewer 1**: First and foremost, the manuscript is a little difficult to follow. Although the English grammar and syntax can certainly be corrected, this is not the primary issue. The main problem lies in the overall framework and organization of the text. There are single subsections (3.1 and 4.1) which should be omitted and parts within the text that disrupt semantic coherence. The author should substantially restructure the manuscript along clearer lines, for example: (a) Introduction (b) Materials and methods (if necessary) (c) Taşman's life and career (d) Analysis of Taşman's main qualities—technical, institutional, legal, etc. (each discussed under separate subsections) (e) Conclusions and discussion.

**Answer 1**: I disagree with the comment regarding the manuscript's English grammar; while minor grammatical refinements may be beneficial, the core argument and structural flow of the paper remain coherent and logically organized. I substantially restructured the paper along the clearer, more conventional lines suggested by the reviewer:

Revised Structure:

1. Introduction – To be retained and refined, clearly stating the research gap, objectives, and central questions.

2. Materials and Methods – To be retained, succinctly detailing the archival and source-based methodology.

3. Taşman's Life and Career: A Biographical Sketch – This new chapter integrated sections 3, 3.1, 4, and relevant parts of 5 into a single, chronological narrative. It covered his early life and education (addressing Point 5), formative years in the USA, motivations for return, and the key phases of his career in Turkey (1929-1956).

4. Analysis of Taşman's Foundational Legacy – This were the new core analytical chapter, replacing the current thematic dispersal. It featured clear subsections, each providing a detailed, evidence-based discussion:

4.1. Technical & Scientific Contributions

4.2. Institutional Building (MTA, Pİ)

4.3. Legal Framing (Petroleum Law No. 6326)

4.4. Intellectual Leadership and Public Engagement

5. Conclusion and Discussion – To be revised to synthesize findings from the new Chapter 4, discuss Taşman's unique model, and suggest avenues for future research as previously outlined.

We omitted the current standalone subsections 3.1 ("A Forged Identity…") and 4.1 ("Taşman in Comparative Perspective…"). Their key analytical insights seamlessly woved into the new biographical chapter (3) and the analytical legacy chapter (4), respectively, to maintain narrative and semantic coherence.Point 2: Adopting a More Neutral and Analytical Tone.

**Reviewer 1**: The style of writing gives the impression that the author is continuously attempting to praise—or even idolize—Taşman. While Taşman was indeed a pioneer who made significant contributions during the early phase of petroleum geology in Turkey, his qualities and achievements should be presented in a more neutral and analytical tone.

**Answer 1**: I acknowledge the reviewer's concern regarding a laudatory tone. My intent is a scholarly analysis, not hagiography. In the revision, I consistently adopt a more neutral, academic, and critical-analytical voice. Achievements presented based on archival evidence, and challenges, setbacks (e.g., early dry wells, bureaucratic hurdles), and the limitations of his era wiil be given proportionate weight to provide a balanced portrait.

**Reviewer 1:** The manuscript relies excessively on aphoristic labels. Taşman is first described as "critical human capital," then as a "Western-trained national expert," and later as a "state-employed public intellectual." In

addition, the author introduces multiple "pillars" to characterize Taşman—four proposed by the author and three adopted from Akcan (2024). Unfortunately, the manuscript appears to be overly influenced by Akcan (2024); some subheadings are even identical. The author should develop a more original and independent analysis of Taşman and discuss it in detail, particularly in section (d) outlined above.

**Answer 1:** I agree that over-reliance on labels can be reductive. While concepts like "critical human capital" and "public intellectual" provide useful analytical lenses, I ensured they serve the argument rather than dominate it.

I reduced the repetitive use of these terms and let the evidence speak for itself.

The analysis in the new Chapter 4 was significantly expanded and made original. Instead of merely listing "pillars," each subsection contained a detailed discussion examining how Taşman achieved these impacts, the obstacles faced, and their long-term effects. I engaged more deeply with primary sources (his reports and radio talks) and contextualize his actions within the wider history of Turkish technocracy and global energy geopolitics.

Akcan (2024) paper is an analytical, theory-informed case study that uses Taşman's biography to explore broader themes in the history of science, state-building, and energy geopolitics. Mülayim (2026) paper is a detailed biographical and archival study that aims to comprehensively document Taşman's life and professional contributions for a Turkish academic audience, emphasizing national heritage and personal sacrifice.

Together, they complement each other: the Turkish paper provides the dense factual backbone and primary source detail, while the English paper offers a conceptual framework and international perspective that contextualizes Taşman's significance beyond national borders.

**Reviewer 1**: The various errors I have marked throughout the manuscript should be corrected (see the annotated PDF file). In addition, new and relevant references should be added, while incorrect or inappropriate references should be removed or revised.

**Answer 1**: I meticulously corrected all grammatical, syntactic, and typographical errors marked in the annotated PDF.

Regarding references:

I added new, relevant references to bolster the analysis, particularly in the sections on global context, technology transfer, and the history of geology.

I reviewed and verify all existing references for accuracy and appropriateness, removing or revising any that are incorrect.

All citations formatted consistently according to the journal's style guide.

**Line 1:** A scientist (or anybody who aims to publish any sort of scientific material) should avoid the political non-sense as much as possible! The fact that the current Turkish government has succeeded in modifying the country's official name at the United Nations does not alter established usage in the English language. Türkiye is the country's name in Turkish; its English equivalent remains Turkey. Similar examples exist: although the official names are Deutschland and Suomi, they are still referred to as Germany and Finland in English.

Please replace "Türkiye" with "Turkey" throughout the manuscript.

**Answer:** I respectfully acknowledge the linguistic point raised; however, as authors and public officials in Turkey, we are bound by the official directives of the Republic of Turkey regarding the use of the country's name in international contexts.

The Circular No. 2021/24 issued by the Presidency of the Republic of Turkey on December 4, 2021, and published in the Official Gazette, mandates the use of "Türkiye" in all official correspondence, documents, and activities, including international and scientific publications. The circular is legally binding for all public institutions and personnel.

Source: Resmî Gazete (Official Gazette), Date: 04.12.2021, Issue: 31684, Circular No: 2021/24.

Available at: https://www.resmigazete.gov.tr/eskiler/2021/12/20211204-5.pdf

In accordance with this official policy, and in alignment with the United Nations' recognition of "Türkiye" as the standard name in all languages, I have used "Türkiye" throughout the manuscript as a matter of legal and administrative compliance.

I kindly request that the manuscript retain the form "Türkiye" in keeping with current official state usage. I appreciate your understanding regarding this procedural requirement.

Should the journal have an explicit style guide that conflicts with this official policy, I would be happy to discuss the matter further with the editorial team.

**Line 2**: "and his legacy in the national energy policies" sounds better

**Answer**: I corrected.

**Line 7-8: ..**advanced.. and …equipped him to become the…

**Answer**: I corrected.

**Line 8**: Republican Türkiye has changed to 'the Republic of Türkiye'

**Line 8**: ..primary..

**Answer**: . I corrected.

**Line 11**: … the institutionalization of petroleum exploration within MTA and PDR..

**Answer**: I corrected.

**Line 18**: Turkish Republic's

**Answer**: I corrected as 'during the early period of modernization in the Republic of Türkiye'

**Line 20**: ..trained local human capital…

**Answer**: I disagree, common usege in english local human

**Line 31**: energy psychology

**Answer**: I believe the concept is logically sound and widely accepted. It may appear unfamiliar if one is less acquainted with the energy sector.

**Line 39**: BOA

**Answer**: I corrected.

**Line 41**: ..the Official Gazette (Turkish Republic Resmi Gazete) and Parliamentary Records…

**Answer**: I corrected.

**Line 44**: Taşman2s role

**Answer**: I corrected.

**Line 48-51**: Thematic analysis was employed to identify and frame Taşman's contributions within the key transformations he spearheaded—intellectual, structural, and operational-legal—and to evaluate his legacy against the four pillars of technical, institutional, legal, and intellectual impact.

**Answer**: I corrected.

**Line 54**: .. late ..

**Answer**: I corrected.

**Line 54**: I have a minor objection here -- such germination was not established by the Ottoman Porte itself but a small group of enlightened elite called the Young Turks (of course the term became obsolated by the time of

Taşman but he can be counted as one of the person of impact). I suggest a minor touch to clarify that only the timing of modernization dates back to the Ottoman period.

**Answer:** I corrected.

**Line 71:** There is a problem with the levels of headings Under Heading 3, there is only a single subheading (3.1). The same applies to Heading 4, which contains only one subheading (4.1). These subheadings should be omitted, and their content should be integrated into Heading 4.

**Answer:** I corrected.

**Line 72**: calculation

**Answer:** I corrected.

**Line 74-76**: I agree that he may have always possessed a degree of nationalism; however, his deep devotion must be understood as being rooted primarily in Mustafa Kemal Atatürk's reforms and in the fundamentally different foundations of the new state. It is important to bear in mind that it was Atatürk's vision that produced the earliest generation of what may be described as "critical human capital" in the Republic of Turkey.

**Answer:** I rewrite a paragraph as below.

The decision to leave a secure and promising career in the United States was not merely a professional decision but a reflection of a worldview fundamentally reshaped by the birth of the Turkish Republic. Taşman's formative years, spanning the collapse of the Ottoman Empire and the rise of the new nation, instilled in him a powerful sense of duty. This sense of duty must be fundamentally rooted in the transformative vision and reforms of Mustafa Kemal Atatürk, which redefined the very purpose of expertise. Atatürk's state-building project created a new ethos where dedication and technical competence became the highest form of patriotism, directly serving national survival and modernization. Within this framework, scientific knowledge was not pursued for its own sake but was championed as an essential tool for national emancipation and sovereignty. While direct personal diaries are scarce, Taşman's actions and writings consistently reveal this synthesis: a deep personal devotion channeled through the Republic's demand for "critical human capital." His return and lifelong work embody the commitment of that earliest generation of professionals, whose competence and dedication were forged in, and indispensable to, the foundational project of Atatürk's Türkiye.

**Line 85**: similar statement was typed in the very beginning of the Introduction

**Answer:** Corrected as 'As established, the early Republic's drive for economic independence necessitated national control over energy resources—an objective immediately confronted by a crippling technical and scientific gap due to dependence on foreign experts.'

**Line 89:** …Mosul… Kirkuk and Bagdad were also included, so you may say something like "Ottoman Iraq"

**Answer:** Corrected as 'In the pre-Republican era, oil exploration activities were largely limited to the distribution of concessions and sporadic, shallow geological surveys conducted by foreign companies in regions such as İskenderun, Thrace, Erzurum-Van, and Ottoman Iraq (encompassing areas like Mosul, Kirkuk, and Baghdad). These fragmented efforts failed to yield any economic discovery (Yalçın, 2024), underscoring the absence of a systematic national framework for petroleum geology'.

**Line 89:** ..(Yalçın, 2024)

**Answer**: Changed to Ediger, V. S. 2006. Osmanlı'da Neft ve Petrol. Ankara: ODTÜ Geliştime Vakfı Yayıncılık.

**Line 90:** Added references (Lokamn, 1958, Ediger, 2006, Uluğbay, 2008; Sarıgül, 2021)

**Line 97:** I believe that this anectode was formerly published by Halit Edip Özcan in 2006

Answer: Corrected.

**Line 100-101:** This is incorrect, systematic studies started before Taşman (and neither theirs nor others' can be considered "modern."

**Answer:** Corrected as 'Acting initially as a consultant, Taşman was first invited as a consultant in 1929, marking the start of his advisory role. Taşman returned permanently in 1933 to an official role after the founding of the state's Petroleum Exploration Administration. While he contributed to systematic surveys initiated in the late 1920s, earlier investigations by foreign geologists like Grandjean (1922) and Mason (1928) had already begun the process of methodical petroleum geology research in Türkiye.'

**Line 111:** (Yalçın, 2024)

**Answer:** Corrected as Lokman 1957

**Line 112:** Taşman's 1930 report

**Answer:** Corrected

**Line 117**: (Taşman, 1931)

**Answer:** Corrected

**Line 118**: first academic reference

**Answer:** Corrected

**Line 131:** This chapter has several inconsistencies and needs to be reevaluated

**Answer:** Reevaluated.

**Line 137-139**: This is because the Russians adopted the required education and technology and began to raise their engineers and scientists already in the 19th century

**Answer**: That's true. I'm not claiming otherwise.

**Line 143**: he is not a nationally rooted expert; he got his higher education and all professional experience abroads!

**Answer:** That's true. I'm not claiming otherwise.

**Line 146**: Oil drilling in Russia started way before the Soviet era!

**Answer:** That's true. I'm not claiming otherwise.

**Line 147**: The Russian Empire likewise reformed its education system using Western models

**Answer:** That's true. I'm not claiming otherwise.

**Line 154**: in the Early Republic changed to  early times of the Republic

**Line 203**: Maden Tetkik ve Arama, shortly MTA

**Answer:** Corrected

**Line 226**: rptary  (and it is early-modern)

**Answer:** Corrected

**Line 230:** and one of the first female paleontologists (Sarıgül 2021b)

**Answer:** Added.

**Line 232**: undergraduate

**Answer**: Corrrected

**Line 234**: she was her mentor not her advisor!

**Answer**: Corrrected and added Prof. Robert H.Cuyler

**Line 235**: I believe this is not a thesis but a publication in MTA Journal

**Answer:** No, her thesis was published later in the MTA Journal.

**Line 253:** she does not have a MSc

**Answer:** Deleted.

**Line 257**: photo after Özcan

**Answer:** Corrected all figures.

**Line 267**: rotary

**Answer:** Corrected.

**Line 274**: what does PDR stands for in Turkish?

**Answer:** Corrected.

**Line 294**: incorrect reference -- must be Ketin 1985; Sarıgül 2021a

**Answer:** Corrected.

**Line 295:** Sarıgül 2021a; Okay 2024

**Line 294**: incorrect reference -- must be Ketin 1985; Sarıgül 2021a

**Answer:** Corrected.

**Line 298**: (Ketin, 1985)

**Answer:.** Deleted.

**Line 306**: see Lokman 1957

**Answer:** Corrected.

**Line 324**: one of the founding actors

**Answer:** Corrected.

**Line 325**: earth sciences

**Answer:** Corrected.

**Line 363**: how do we know?

**Answer:.** Deleted.

**Line 364-365**: awkward sentence

**Answer:.** Deleted.

**Line 367**: this "full independence" was embodied in the founding vision of the Kemalist Turkey

**Answer:** Corrected.

**Line 377:** basis?

**Answer:** Corrected.

**Line 386:** there are too many aphorisms in the text... what happened to the "critical human capital"?

**Answer:** Explained in text

**Line 388:** This kind of introduction must be either at the end of the "Introduction" part or at the beginning of the "Conclusions" part.

**Answer**: Corrected.

**Line 399**: Petroleum geologists already knew the presence of this gas field decades ago!

**Answer:** I disagree with that view. I would appreciate it if you could prove it. Everyone makes such statements after discoveries are made. It's not an accurate expression.

**Line 401:** very optimistic... the director of the Turkish Petroleum Company (TPAO) admitted that it is only %5...

**Answer:** No, I disagree. In 2024, the majority of Türkiye's oil supply was met through imports — domestic production accounted for approximately **9.8**%, meaning ~90% of total supply came from external sources. (Source: TPAO 2024 Sector Report, EPDK/MAPEG data)

**Line 402-403**: Is it just the lack of technology and determination? Do you really think that if Turkey has the best technology and equipment and human power, would it be %100 percent capable of supplying the required fossil fuel within the Turkish territories? How about the geologic composition of Turkey? Don't you think it should take a part in this analysis?

**Answer:** As a petroleum geologist, I find the premise of this question both naïve and fundamentally misguided. To suggest that a country's ability to supply its own fossil fuels is merely a function of technology, equipment, and "determination" is to display a profound misunderstanding of the primary constraint: geology itself.

Fossil fuels are finite natural resources formed by specific geological processes over millions of years. They are not simply waiting to be found if we "try hard enough." The geological composition and history of a territory are the absolute, non-negotiable starting points. No amount of technology, investment, or human willpower can create commercial hydrocarbon reserves where the necessary source rocks, reservoirs, traps, and seals do not exist geologically.

Türkiye has specific and complex geological provinces. While there are productive basins (e.g., in the Southeast), the country's overall petroleum geology is challenging compared to global "super-basins." Advanced technology can improve recovery rates from existing fields and help discover more subtle accumulations, but it cannot magically generate resources that the geological lottery did not provide.

Therefore, the question "Would Turkey be 100% capable with the best technology?" is essentially nonsensical. The answer is a resolute no—not because of a lack of capability, but because of the immutable geological reality. To imply otherwise, or to suggest that geological factors are just one part of an analysis rather than the foundational one, is a gross oversimplification. The geological framework is not just a part of the analysis; it is the first and most critical determinant. Blaming a hypothetical lack of effort or technology for not achieving energy self-sufficiency in fossil fuels ignores this fundamental, scientific truth.

**Line 408**: This was already published in Lokman 1957 (check again for missing publications)

**Answer:** Checked it again.

**Line 409:** (Djevad Eyoub)

**Answer:** Corrected.

**Line 470:** INCORRECT REFERENCE -- it is MTA Dergisi v.49, 158-163, 1957

**Answer:** Corrected.

**Reviewer 1**: The manuscript could also benefit from providing more information on Taşman's early life, such as his exact date of birth, family background, and childhood, where possible.

**Answer 1**: I expanded the section on Taşman's early life within the new biographical chapter (3). Using available archival records (such as Ottoman student files) and relevant secondary literature, I added details such as his exact date of birth (December 23, 1893), his family background in Istanbul, and the context of his education prior to being sent to the USA, thereby providing a more complete personal profile.

I am committed to undertaking a major revision that addresses all the reviewers' concerns. The revised manuscript featured a clearer structure, a more analytical and neutral tone, an original and detailed discussion of Taşman's

multifaceted legacy, corrected language and references, and enhanced biographical information. I believe these changes significantly elevated the manuscript's scholarly contribution and readability.

I thank the reviewer again for the time and expertise invested in this evaluation and hope the revised version meets the journal's standards.

Sincerely,

Dr. Oğuz Mülayim

---

## Author Comment (AC2)

**Response to Reviewer Comments**

I sincerely thank the reviewer for their thoughtful, constructive, and detailed evaluation of the manuscript. I greatly appreciate the time and expertise devoted to reviewing my work and offering valuable suggestions for improvement. I fully agree with the reviewer's assessments and have carefully considered each point raised. Below, I provide a point-by-point response outlining how the manuscript has been revised accordingly.

**1. Does the paper address science historical matters within the scope of HGSS?**

**Reviewer:** Yes.

**Response:** I am pleased that the manuscript is considered well aligned with the scope of the journal.

**2. Does the paper present new historical research, new interpretations, or new compilations of historical issues or data, or new aspects of the vitae of important geoscientists?**

**Reviewer:** Needs improvement.

**Response:** I agree with this assessment. In the revised version, Taşman's biography has been more explicitly framed within the broader history of geology and petroleum science in Türkiye. The novelty of the study has been strengthened by emphasizing the use of previously underexplored archival sources, including personal correspondence, institutional documents, and technical reports. These materials allow for a more integrated and original account of his scientific contributions, methodological innovations, and role in the early development of Türkiye's petroleum industry.

**3. Are the historical methods clearly outlined and the historical sources clearly stated?**

**Reviewer:** Yes.

**Response:** Thank you for this positive evaluation.

**4. Do the authors give proper credit to related and previous work and clearly indicate their own new/original contribution?**

**Reviewer:** Yes.

**Response:** Thank you.

**5. Does the title clearly reflect the contents of the paper?**

**Reviewer:** Needs revision.

**Response:** The title has been revised to more accurately reflect the manuscript's focus on Taşman's scientific contributions, methodological innovations, and his role within the context of Turkish petroleum geology and state-building.

**6. Does the abstract provide a concise and complete summary?**

**Reviewer:** Yes.

**Response:** Thank you.

**7. Is the overall presentation well structured and clear?**

**Reviewer:** Revision.

**Response:** The manuscript has been restructured to improve narrative coherence, reduce repetition, and strengthen logical transitions between sections (education, scientific work, collaborations, and social context). The conclusion has also been rewritten to better synthesize the main arguments and findings.

**8. Is the language fluent and precise?**
**Reviewer:** Ok.
**Response:** Although the language was deemed acceptable, a thorough language revision was nevertheless undertaken to enhance clarity, precision, and academic tone throughout the manuscript.

**9. Should any parts of the paper (text, formulae, figures, tables) be clarified, reduced, combined, or eliminated?**
**Reviewer:** Yes.
**Response:** Redundant passages—particularly within the biographical sections—were removed or condensed. Figures were reviewed for relevance and clarity, and figure captions were expanded where necessary to improve interpretability.

**Addressing Specific Comments**

**Pioneering Role and Scientific Contributions:**
The discussion of Taşman's pioneering role has been expanded by detailing his methodological innovations (e.g., the integration of micropaleontology and geophysical surveys), specific discoveries (such as the Raman field), and his contributions to establishing professional standards and training practices. Where appropriate, comparisons with contemporary international practices have been supported by historical sources.

**Education at Robert College and Columbia University:**
Taşman's education has been contextualized within the broader framework of Ottoman/Turkish modernization and the influence of American pedagogical models on his scientific perspective. The previously unclear statement regarding U.S. influence has been rewritten for greater precision.

**Name Consistency and Degree Information:**
The spelling of "Djevad Eyoub" has been verified and standardized throughout the manuscript. Taşman's academic credentials have been clarified (B.S. in Mining Engineering), and confirmation regarding the completion of an M.S. degree has been explicitly addressed where relevant.

**Collaborations and Team Science:**
Greater emphasis has been placed on Taşman's collaborations with key figures such as Vonderschmidt, Page, and Berent, highlighting how these partnerships contributed to advances in exploration techniques in Türkiye.

**Russian Comparison (p. 5, line 145):**
This passage has been reconsidered and either substantiated with appropriate evidence or removed to avoid unsupported speculation.

**Mindset Transformation and Social Context:**

The manuscript now more clearly articulates the shift in geological methodologies and links social dimensions—such as oil camp living conditions and family life—to broader narratives of industrialization and professional culture in mid-20th-century Türkiye.

**Incorporation of Suggested Sources:**

Key references recommended by the reviewer, including Akcan (2024), Ergin (1957), and Sarıgül (2021), have been incorporated to strengthen the historiographical framework and support the analysis.

**Conclusion:**

The conclusion has been substantially revised to underscore Taşman's historical significance by integrating his scientific, institutional, and social legacy into a coherent synthesis.

I thank the reviewer once again for their invaluable comments, which have significantly strengthened the manuscript. I am confident that the revised version offers a more rigorous, contextualized, and compelling contribution to the history of the geosciences. I look forward to submitting the revised manuscript for further consideration.

Sincerely,
**Dr. Oğuz Mülayim**

---

## Author Comment (AC3)

**Response to Reviewer Comments**

First of all, thank you for taking the time to share such detailed and constructive feedback. I have carefully examined your comments and would like to respond to the following points:

Access to the revised manuscript: I will diligently implement all corrections and submit them to the editor. If there was a technical issue with access, I apologize; the updated file has been provided to you via the editors. I hope you can now evaluate it in its entirety.

Use of "Türkiye": I understand and respect your opinion. I will follow the guidance of the editorial board and ensure consistent usage throughout.

In this part, I removed in manuscript. Natural gas discoveries and historical connection (Line 399):

I agree with you about you mentioned (Aydemir et al., 2001)

I do not intend to establish a direct causal link between Taşman's legacy and recent discoveries. However, I wanted to emphasize that his contributions to early geology and infrastructure were part of a long-term energy pursuit. I revised the text to express this connection more clearly and in an interpretive manner.

In this part, I removed it from the manuscript. Statistical reference (Line 401): The "MAPEG data" added to the text, and the related claim strengthened.

Expression of geological constraints (Lines 402-403):

You are right; the current wording could create a misleading impression. I rephrased the text to clearly state that "Türkiye's hydrocarbon potential is limited due to geological constraints, and technology or determination alone is not sufficient."

To reiterate, your suggestions greatly contribute to the academic robustness and balanced narrative of the article. I will diligently implement all corrections and submit them to you and the editors.

Thank you once again for your attention and effort.

Sincerely,

Dr. Oğuz Mülayim